# Learning to Ground Multi-Agent Communication with Autoencoders

**Toru Lin**
MIT CSAIL
torulk@mit.edu

**Minyoung Huh**
MIT CSAIL
minhuh@mit.edu

**Chris Stauffer**
Facebook AI
cstauffer@fb.com

**Ser-Nam Lim**
Facebook AI
sernamlim@fb.com

**Phillip Isola**
MIT CSAIL
phillipi@mit.edu

## Abstract

Communication requires having a common language, a lingua franca, between agents. This language could emerge via a consensus process, but it may require many generations of trial and error. Alternatively, the lingua franca can be given by the environment, where agents ground their language in representations of the observed world. We demonstrate a simple way to ground language in learned representations, which facilitates decentralized multi-agent communication and coordination. We find that a standard representation learning algorithm – autoencoding – is sufficient for arriving at a grounded common language. When agents broadcast these representations, they learn to understand and respond to each other's utterances and achieve surprisingly strong task performance across a variety of multi-agent communication environments.

## 1 Introduction

An essential aspect of communication is that each pair of speaker and listener must share a common understanding of the symbols being used [8]. For artificial agents interacting in an environment, with a communication channel but without an agreed-upon communication protocol, this raises the question: *how can meaningful communication emerge before a common language has been established?*

To address this challenge, prior works have used supervised learning [18], centralized learning [14, 17, 28], or differentiable communication [6, 17, 28, 32, 41]. Yet, none of these mechanisms is representative of how communication emerges in nature, where animals and humans have evolved communication protocols without supervision and without a centralized coordinator [35]. The communication model that most closely resembles language learning in nature is a fully decentralized model, where agents' policies are independently optimized. However, decentralized models perform poorly even in simple communication tasks [17] or with additional inductive biases [10].

We tackle this challenge by first making the following observations on why emergent communication is difficult in a decentralized multi-agent reinforcement learning setting. A key problem that prevents agents from learning meaningful communication is the lack of a common grounding in communication symbols [3, 10, 19]. In nature, the emergence of a common language is thought to be aided by physical biases and embodiment [29, 42] – we can only produce certain vocalizations, these sounds only can be heard a certain distance away, these sounds bear similarity to natural sounds in the environment, etc. – yet artificial communication protocols are not a priori grounded in aspects of the environment

---

Project page, code, and videos can be found at https://toruowo.github.io/marl-ae-comm/.

35th Conference on Neural Information Processing Systems (NeurIPS 2021).

dynamics. This poses a severe exploration problem as the chances of a consistent protocol being found and rewarded is extremely small [17]. Moreover, before a communication protocol is found, the random utterances transmitted between agents add to the already high variance of multi-agent reinforcement learning, making the learning problem even more challenging [10, 28].

To overcome the grounding problem, an important question to ask is: do agents really need to learn language grounding from scratch through random exploration in an environment where success is determined by chance? Perhaps nature has a different answer; previous studies in cognitive science and evolutionary linguistics [20, 36, 39, 40] have provided evidence for the hypothesis that communication first started from sounds whose meaning are grounded in the physical environment, then creatures adapted to make sense of those sounds and make use of them. Inspired by language learning in natural species, we propose a novel framework for grounding multi-agent communication: first ground *speaking* through learned representations of the world, then learn *listening* to interpret these grounded utterances. Surprisingly, even with the simple representation learning task of autoencoding, our approach eases the learning of communication in fully decentralized multi-agent settings and greatly improves agents' performance in multi-agent coordination tasks that are nearly unsolvable without communication.

The contribution of our work can be summarized as follows:

- We formulate communication grounding as a representation learning problem and propose to use observation autoencoding to learn a common grounding across all agents.
- We experimentally validate that this is an effective approach for learning decentralized communication in MARL settings: a communication model trained with a simple autoencoder can consistently outperform baselines across various MARL environments.
- In turn, our work highlights the need to rethink the problem of emergent communication, where we demonstrate the essential need for visual grounding.

## 2 Related Work

In multi-agent reinforcement learning (MARL), achieving successful emergent communication with decentralized training and non-differentiable communication channel is an important yet challenging task that has not been satisfactorily addressed by existing works. Due to the non-stationary and non-Markovian transition dynamics in multi-agent settings, straightforward implementation of standard reinforcement learning methods such as Actor-Critic [22] and DQN [31] perform poorly [17, 28].

Centralized learning is often used to alleviate the problem of high variance in MARL, for example learning a centralized value function that has access to the joint observation of all agents [9, 14, 28]. However, it turns out that MARL models are unable to solve tasks that rely on emergent communication, even with centralized learning and shared policy parameters across the agents [17]. Eccles et al. [10] provides an analysis that illustrates how MARL with communication poses a more difficult exploration problem than standard MARL, which is confirmed by empirical results in [17, 28]: communication exacerbates the sparse and high variance reward signal in MARL.

Many works therefore resort to differentiable communication [6, 17, 28, 32, 41], where agents are allowed to directly optimize each other's communication policies through gradients. Among them, Choi el al. [6] explore a high-level idea that is similar to ours: generating messages that the model itself can interpret. However, these approaches impose a strong constraint on the nature of communication, which limits their applicability to many real-world multi-agent coordination tasks.

Jaques et al. [21] proposes a method that allows independently trained RL agents to communicate and coordinate. However, the proposed method requires that an agent either has access to policies of other agents or stays in close proximity to other agents. These constraints make it difficult for the same method to be applied to a wider range of tasks, such as those in which agents are not embodied or do not observe others directly. Eccles et al. [10] attempts to solve the same issue by introducing inductive biases for positive signaling and positive listening, but implementation requires numerous task-specific hyperparameter tuning, and the effectiveness is limited.

It is also worth noting that, while a large number of existing works on multi-agent communication take structured state information as input [4, 15, 16, 17, 32, 34], we train agents to learn a communication protocol directly from raw pixel observations. This presents additional challenges due to

the unstructured and ungrounded nature of pixel data, as shown in [3, 6, 24]. To our knowledge, this work is the first to effectively use representation learning to aid communication learning from pixel inputs in a wide range of MARL task settings.

## 3 Preliminaries

We model **multi-agent reinforcement learning (MARL) with communication** as a partially-observable general-sum Markov game [26, 38], where each agent can broadcast information to a shared communication channel. Each agent receives a partial observation of the underlying world state at every time step, including all information communicated in the shared channel. This observation is used to learn an appropriate policy that maximizes the agent's environment reward. In this work, we parameterize the policy function using a deep neural network.

Formally, a decentralized MARL can be expressed as a partially observable Markov decision process as $\mathcal{M} = \langle \mathcal{S}, \mathcal{A}, \mathcal{C}, \mathcal{O}, \mathcal{T}, R, \gamma \rangle$, where $N$ is the number of agents, $\mathcal{S}$ is a set of states spaces, $\mathcal{A} = \{\mathcal{A}^1, ..., \mathcal{A}^N\}, \mathcal{C} = \{\mathcal{C}^1, ..., \mathcal{C}^N\}$, and $\mathcal{O} = \{\mathcal{O}^1, ..., \mathcal{O}^N\}$ are a set of action, of communication, and of observation spaces respectively.

At time step $t$, an agent $k$ observes a partial view $o_t^{(k)}$ of the underlying true state $s_t$, and a set of communicated messages from the previous time step $c_{t-1} = \{c_{t-1}^{(1)}, ..., c_{t-1}^{(N)}\}$. The agent then chooses an action $a_t^{(k)} \in \mathcal{A}^k$ and a subsequent message to broadcast $c_t^{(k)} \in \mathcal{C}^k$. Given the joint actions of all $N$ agents $a_t = \{a_t^{(1)}, ..., a_t^{(N)}\} \in (\mathcal{A}^1, ..., \mathcal{A}^N)$, the transition function $\mathcal{T} : \mathcal{S} \times \mathcal{A}^1 \times ... \times \mathcal{A}^N \to \Delta(\mathcal{S})$ maps the current state $s_t$ and set of agent actions $a_t$ to a distribution over the next state $s_{t+1}$. Since the transition function $\mathcal{T}$ is non-determinstic, we denote the probability distributions over $\mathcal{S}$ as $\Delta(\mathcal{S})$. Finally, each agent receives an individual reward $r_t^{(k)} \in R(s_t, a_t)$ where $R : \mathcal{S} \times \mathcal{A}^1 \times ... \times \mathcal{A}^N \to \mathbb{R}$.

In our work, we consider a fully cooperative setting in which the objective of each agent is to maximize the total expected return of all agents:

$$\underset{\pi:\mathcal{S}\to\mathcal{A}\times\mathcal{C}}{\text{maximize}} \quad \mathbb{E}\Big[ \sum_{t\in T}\sum_{k\in N} \gamma^t R(s_t, a_t) \,\Big|\, (a_t, c_t) \sim \pi^{(k)}, s_t \sim \mathcal{T}(s_{t-1}) \Big] \tag{1}$$

for some finite time horizon $T$ and discount factor $\gamma$.

In MARL, the aforementioned objective function is optimized using policy gradient. Specifically, we use asynchronous advantage actor-critic (A3C) [30] with Generalized Advantage Estimation [37] to optimize our policy network. The policy network in A3C outputs a distribution over the actions and the discounted future returns. Any other policy optimization algorithms could have been used in lieu of A3C. We do not use centralized training or self-play, and only consider decentralized training where each agent is parameterized by an independent policy.

## 4 Grounding Representation for Communication with Autoencoders

The main challenge of learning to communicate in fully decentralized MARL settings is that there is no grounded information to which agents can associate their symbolic utterances. This lack of grounding creates a dissonance across agents and poses a difficult exploration problem. Ultimately, the gradient signals received by agents are therefore largely inconsistent. As the time horizon, communication space, and the number of agents grow, this grounding problem becomes even more pronounced. This difficulty is highlighted in numerous prior works, with empirical results showing that agents often fail to use the communication channel at all during decentralized learning [3, 10, 19, 28].

We propose a simple and surprisingly effective approach to mitigate this issue: using a self-supervised representation learning task to learn a common grounding [43] across all agents. Having such a grounding enables speakers to communicate messages that are understandable to the listeners and convey meaningful information about entities in the environment, though agents need not use the same symbols to mean the same things. Specifically, we train each agent to independently learn to auto-encode its own observation and use the learned representation for communication. This approach offers the benefits of allowing fully decentralized training without needing additional architectural

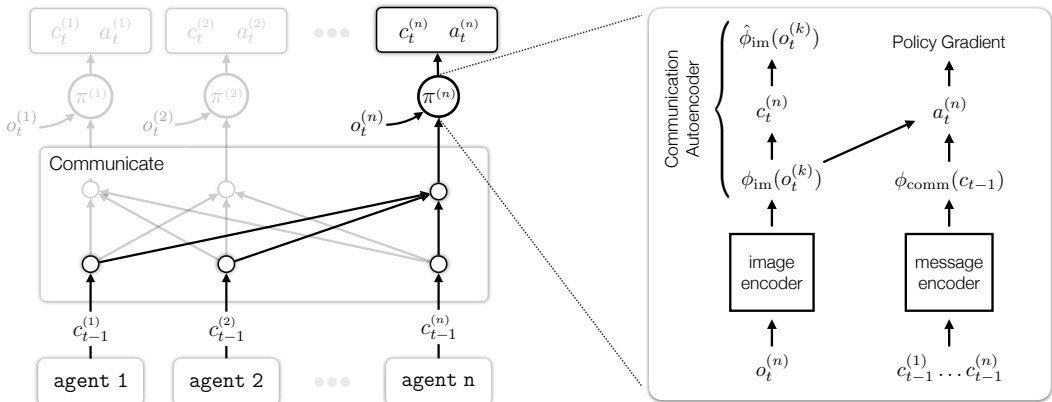

Figure 1: **Overview:** The overall schematic of our multi-agent system. All agents share the same individual model architecture, but each agent is independently trained to learn to auto-encode its own observation and use the learned representation for communication. At each time step, each agent observes an image representation of the environment as well as messages broadcasted by other agents during the last time step. The image pixels are processed through an Image Encoder; the broadcasted messages are processed through a Message Encoder; the image features and the message features are concatenated and passed through a Policy Network to predict the next action. The image features are also used to generate the next communication messages using the Communication Autoencoder.

bias or supervision signal. In Section 5, we show the effectiveness of our approach on a variety of MARL communication tasks.

An overview of our method is shown in Figure 1, which illustrates the communication flow among agents at some arbitrary time step $t$. All agents share the same individual model architecture, and each agent consists of two modules: a **speaker module** and a **listener module**. We describe the architecture details of a single agent $k$ below.

## 4.1 Speaker Module

At each time step $t$, the speaker module takes in the agent's observation $o_t^{(k)}$ and outputs the agent's next communication message $c_t^{(k)}$.

**Image Encoder** Given the raw pixel observation, the module first uses a image encoder to embed the pixels into a low-dimensional feature $o_t^{(k)} \rightarrow \phi_{\text{im}}(o_t^{(k)}) \in \mathbb{R}^{128}$. The image encoder is a convolutional neural network with $4$ convolutional layers, and the output of this network is spatially pooled. We use the same image encoder in the listener module.

**Communication Autoencoder** The goal of the communication autoencoder is to take the current state observation and generate the next subsequent message. We use an autoencoder to learn a mapping from the feature space of image encoder to communication symbols, i.e. $\phi_{\text{im}}(o_t^{(k)}) \rightarrow c_t^{(k)}$. The autoencoder consists of an encoder and a decoder, both parameterized by a 3-layer MLP. The decoder tries to reconstruct the input state from the communication message $c_t^{(k)} \rightarrow \hat{\phi}_{\text{im}}(o_t^{(k)})$. The communication messages are quantized before being passed through the decoder. We use a straight-through estimator to differentiate through the quantization [1]. The auxiliary objective function of our model is to minimize the reconstruction loss $\|\phi_{\text{im}}(o_t^{(k)}) - \hat{\phi}_{\text{im}}(o_t^{(k)})\|_2^2$. This loss is optimized jointly with the policy gradient loss from the listener module.

## 4.2 Listener Module

While the goal of the speaker module is to output grounded communication based on the agent's private observation $o_t^{(k)}$, the goal of the listener module is to learn an optimal action policy based on both the observation $o_t^{(k)}$ and communicated messages $c_{t-1}$. At each time step $t$, the listener module outputs the agent's next action $a_t^{(k)}$.

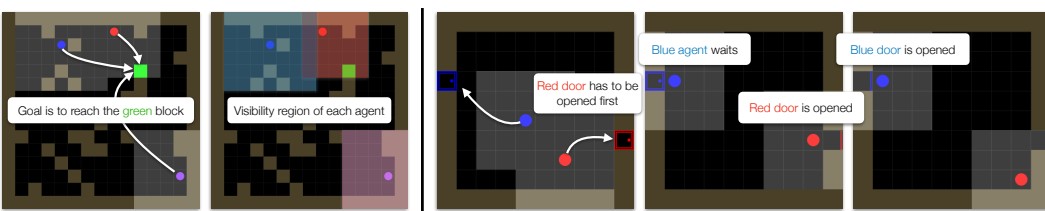

Figure 2: **MarlGrid Environment:** We introduce two new grid environments: `FindGoal` (left) and `RedBlueDoors` (right). These environments are adapted from the GridWorld environment [5, 33]. Environment states are randomized at every episode and are partially observable to the agents. In `FindGoal`, the task is to reach the green goal location. Each agent receives a reward of $1$ when they reach the goal, and an additional reward of $1$ when all 3 agents reach the goal within the time frame. In `RedBlueDoors`, the task is ordinal, where the ordering of actions matter. A reward of $1$ is given to both agents if and only if the red door is opened first and then the blue door.

**Message Encoder**    The message encoder linearly projects all messages communicated from the previous time step $c_{t-1}$ using a shared embedding layer. The information across all agent message embeddings is combined through concatenation and passed through 3-layer MLP. The resulting message feature has a fixed dimension of 128, i.e. $\phi_{\text{comm}}(c_t) \in \mathbb{R}^{128}$.

**Policy Network**    Each agent uses an independent policy head, which is a standard GRU [7] policy with a linear layer. The GRU policy concatenates the encoded image features and the message features $\phi_t^{(k)} = \phi_{\text{im}}(o_t^{(k)}) \circ \phi_{\text{comm}}(c_t)$, where $\circ$ is the concatenation operator across the feature dimension. The GRU policy predicts a distribution over the actions $a \sim \pi(\phi_t^{(k)})$ and the corresponding expected returns. The predicted action distribution and expected returns are used for computing the policy gradient loss. This loss is jointly optimized with the autoencoder reconstruction loss from the speaker module.

The same setup is used for all experiments. The exact details of the network architecture and the corresponding training details are in the Appendix.

## 5 Experiments

In this section, we demonstrate that autoencoding is a simple and effective representation learning algorithm to ground communication in MARL. We evaluate our method on various multi-agent environments and qualitatively show that our method outperforms baseline methods. We then provide further ablations and analyses on the learned communication.

### 5.1 Environments

We introduce three multi-agent communication environments: `CIFAR Game`, `FindGoal`, and `RedBlueDoors`. Our work focuses on fully cooperative scenarios, but can also be extended to competitive or mixed scenarios.

Our environments cover a wide range of communication task settings, including (1) referential or non-referential, (2) ordinal or non-ordinal, and (3) two-agent versus generalized multi-agent. A referential game, often credited to Lewis signaling game [25], refers to a setup in which agents communicate through a series of message exchanges to solve a task. In contrast to non-referential games, constructing a communication protocol is critical to solving the task – where one can only arrive at a solution through communication. Referential games are referred to as a grounded learning environment, and therefore, communication in MARL has been studied mainly through the lens of referential games [12, 24]. Lastly, ordinal tasks refer to a family of problems where the ordering of the actions is critical for solving the task. The difference between ordinal and non-ordinal settings is illustrated in Figure 2, where the blue agent must wait for the red agent to open the door to successfully receive a reward. In contrast, non-ordinal tasks could benefit from shared information, but it is not necessary to complete the task. We now describe the environments used in our work in more detail:

**CIFAR Game**   We design `CIFAR Game` following the setup of `Multi-Step MNIST Game` in [17], but with `CIFAR-10` dataset [23] instead. This is a non-ordinal, two-agent, referential game. In `CIFAR` game, each agent independently observes a randomly drawn image from the `CIFAR-10` dataset, and the goal is to communicate the observed image to the other agent within 5 environment time steps. At each time step, each agent broadcasts a set of communication symbols of length $l$. At the final time step, each agent must choose a class label from the 10 possible choices. At the end of the episode, an agent receives a reward of 0.5 for each correctly guessed class label, and both agents receive a reward of 1 only when both images are classified correctly.

**MarlGrid Environments**   The second and third environments we consider are: `FindGoal` (Figure 2 left) and `RedBlueDoors` (Figure 2 right). Both environments are adapted from the GridWorld environment [5, 33] and environment states are randomized at every episode.

`FindGoal` is a non-ordinal, multi-agent, non-referential game. We use $N = 3$ agents, and at each time step, each agent observes a partial view of the environment centered at its current position. The task of agents is to reach the green goal location as fast as possible. Each agent receives an individual reward of 1 for completing the task and an additional reward of 1 when all agents have reached the goal. Hence, the optimal strategy of an agent is to communicate the goal location once it observes the goal. If all agents learn a sufficiently optimized search algorithm, they can maximize their reward without communication.

`RedBlueDoors` is an ordinal, two-agent, non-referential game. The environment consists of a red door and a blue door, both initially closed. The task of agents is to open both doors, but unlike in the previous two games, the ordering of actions executed by agents matters. A reward of 1 is given to both agents if and only if the red door is opened first and then the blue door. This means that any time the blue door is opened first, both agents receive a reward of 0, and the episode ends immediately. Hence, the optimal strategy for agents is to convey the information that the red door was opened. Since it is possible to solve the task through visual observation or by a single agent that opens both doors, communication is not necessary.

Compared to `CIFAR Game`, the MarlGrid environments have a higher-dimensional observation space and a more complex action space. The fact that these environments are non-referential exacerbates the visual-language grounding problem since communication can only exist in the form of *cheap talk* (i.e., costless, nonbinding, nonverifiable communication [13] that has no direct effect on the game state and agent payoffs). We hope to show from this set of environments that autoencoders can be used as a surprisingly simple and adaptable representation learning task to ground communication. It requires little effort to implement and almost no change across environments. Most importantly, as we will see in Section 5.4, autoencoded representation shows an impressive improvement over communication trained with reinforcement learning.

### 5.2   Baselines

To evaluate the effectiveness of grounded communication, we compare our method (`ae-comm`) against the following baselines: (1) a `no-comm` baseline, where agents are trained without a communication channel; (2) a `rl-comm` baseline*, where we do not make a distinction between the listener module and the speaker module, and the communication policy is learned in a way similar to the environment policy; (3) a `rl-comm-with-biases` baseline, where inductive biases for positive signaling and positive listening are added to `rl-comm` training as proposed in [10]; (4) a `ae-rl-comm` baseline, where the communication policy is learned by an additional policy network trained on top of the autoencoded representation in speaker module.

### 5.3   The Effectiveness of Grounded Communication

In Table 1 and Figure 3, we compare task performance of `ae-comm` agents with performance of baseline agents. We report all results with 95% confidence intervals, evaluated on 100 episodes per seed over 10 random seeds.

In `CIFAR Game` environment, all three baselines (`no-comm`, `rl-comm`, `rl-comm-with-biases`) could only obtain an average reward close to that of random guesses throughout the training process.

---

*The `rl-comm` baseline can be seen as an A3C version of RIAL [17] without parameter sharing.

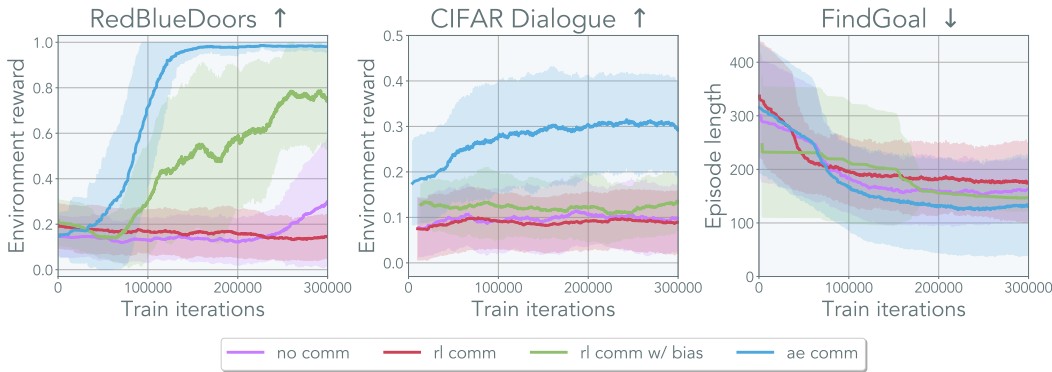

Figure 3: **Comparison with baselines:** Comparison between our method that uses an autoencoded communication (`ae-comm`), a baseline that is trained without communication (`no-comm`), a baseline where communication policy is trained using reinforcement learning (`rl-comm`), and another baseline where inductive biases for positive signaling and positive listening are added to `rl-comm` training (`rl-comm-with-inductive-biases`). For `FindGoal`, we visualize the amount of time it takes for all agents to reach the goal, as all methods can reach the goal within the time frame. For each set of results, we report the mean and 95% confidence intervals evaluated on 100 episodes per seed over 10 random seeds.

In comparison, `ae-comm` agents achieve a much higher reward on average. Since this environment is a referential game, our results directly indicate that `ae-comm` agents learn to communicate more effectively than the baseline agents.

`RedBlueDoors` environment poses a challenging multi-agent coordination problem since the reward is extremely sparse. As shown in Figure 3(b), neither of the `no-comm` and `rl-comm` agents was able to learn a successful coordination strategy. Although `rl-comm-with-biases` agents outperform the other two baseline agents, they do not learn an optimal strategy that guarantees an average reward close to 1. In contrast, `ae-comm` agents converge to an optimal strategy after 150k of training.

| methods | CIFAR | RedBlueDoors | FindGoal |
|---|---|---|---|
| | **avg. $r \uparrow$** | **avg. $r \uparrow$** | **avg. $t \downarrow$** |
| no-comm | $0.082 \pm 0.009$ | $0.123 \pm 0.096$ | $169.0 \pm 26.8$ |
| rl-comm | $0.099 \pm 0.013$ | $0.174 \pm 0.009$ | $184.8 \pm 25.2$ |
| rl-comm w/ bias [10] | $0.142 \pm 0.019$ | $0.729 \pm 0.072$ | $158.0 \pm 12.3$ |
| ae-comm (ours) | $\mathbf{0.348 \pm 0.041}$ | $\mathbf{0.984 \pm 0.002}$ | $\mathbf{103.5 \pm 20.2}$ |

Table 1: **Comparison with baselines:** We compute the average reward for `CIFAR Game` and `RedBlueDoors` environments, and average episode length for `FindGoal` environment. For each set of results, we report the mean and 95% confidence intervals evaluated on 100 episodes per seed over 10 random seeds.

In `FindGoal` environment, agents are able to solve the task without communication, but their performance can be improved with communication. Therefore, we use episode length instead of reward as the performance metric for this environment. To resolve the ambiguity in Figure 3(c), we also include numerical results in a table below. While all agents are able to obtain full rewards, Figure 3 shows that `ae-comm` agents are able to complete the episode much faster than other agents. We further verify that this improvement is indeed a result of the successful communication by providing further analysis in Section 5.5.

Our results indicate that a communication model trained with autoencoding tasks consistently outperforms the baselines across all environments. The observation that communication does not work well with reinforcement learning is consistent with observations made in prior works [10, 17, 28]. Furthermore, our results with autoencoders – a task that is often considered trivial – highlight that we as a community may have overlooked a critical representation learning component in MARL communication. In Section 5.4, we provide a more detailed discussion on the difficulty of training emergent communication via policy optimization.

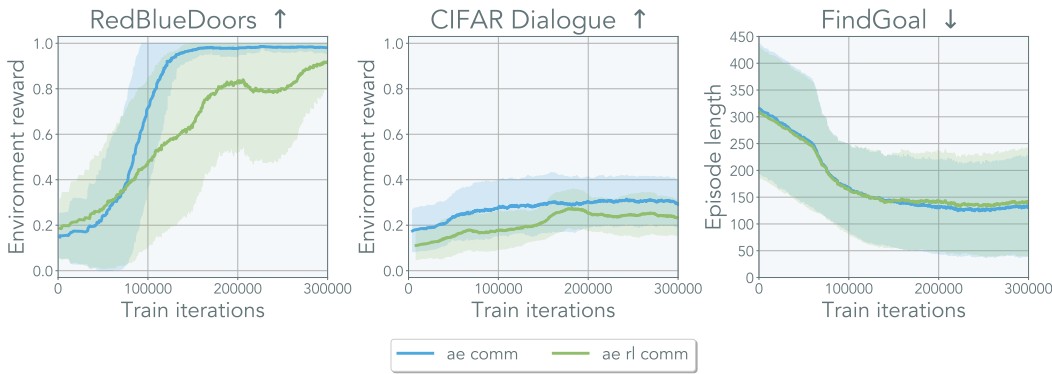

Figure 4: **Representation learning with reinforcement learning:** Comparison between a speaker module trained with only an autoencoding task (`ae-comm`) and another one trained with both autoencoding task and reinforcement learning (`ae-rl-comm`). We observed that further training a policy on top of the autoencoder representation degrades performance across all environments (except for in `FindGoal`, where performance of `ae-rl-comm` and `ae-comm` stayed about the same).

## 5.4 The Role of Autoencoding

Given the success of agents trained with autoencoders, it is natural to ask whether a better communication protocol can emerge from the speaker module by jointly training it with a reinforcement learning policy. To this end, we train a GRU policy on top of the autoencoded representation (`ae-rl-comm`) and compare it against our previous model that was trained just with an autoencoder (`ae-comm`). The communication policy head is independent of the environment action policy head.

Surprisingly, we observed in Figure 4 that the model trained jointly with reinforcement learning consistently performed worse (except for in `FindGoal`, where performance of `ae-rl-comm` and `ae-comm` stayed about the same). We hypothesize that the lack of correlation between visual observation and the communication rewards hurts the agents' performance. This lack of visual-reward grounding could introduce conflicting gradient updates to the action policy, and thereby exacerbate the high-variance problem that already exists in reinforcement learning. In other words, optimization is harder whenever the joint-exploration problem for learning speaker and listener policies is introduced. Our observation suggests that specialized reward design and training at the level of [2] might be required for decentralized MARL communication. This prompts us to rethink how to address the lack of visual grounding in communication policy, where this work serves as a first step in this direction.

## 5.5 Analyzing the Effects of Communication Signals on Agent Behavior

**Communication Embedding**  To analyze whether the agents have learned a meaningful visual grounding for communication, we first visualize the communication embedding. In Figure 5, we visualize the communication symbols transmitted by the agents trained on `RedBlueDoors`. The communication symbols are discrete with a length of $l = 10$ (1024 possible embedding choices), and we use approximately 4096 communication samples across 10 episodic runs. We embed the communication symbols using t-SNE [44] and further cluster them using DBSCAN [11]. In the figure, we visualize clusters by observing the correspondence between image states and communication symbols produced by agents in those states. For example, we observed that a specific communication cluster corresponded to an environment state when the red door was opened; this suggests a communication action where one agent signals the other agent to open the blue door and complete the task.

**Entropy of Action Distribution**  To measure whether communicated information directly influences other agents' actions, we visualize the entropy of action distribution during episode rollouts. Suppose one agent shares information that is vital to solving the task. In that case, a decrease in entropy should be observed in the action distributions of other agents, as they act more deterministically towards solving the task.

As shown in Figure 6, we visualize the entropy of action distribution across 256 random episodic runs using policy parameters from a fully trained `ae-comm` model. The entropies are aligned using

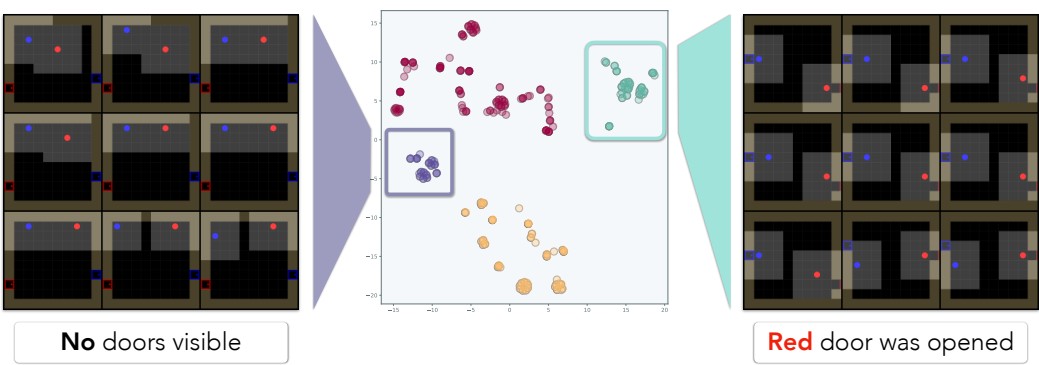

**Communication Embedding**

| **No** doors visible | **Red** door was opened |
|---|---|

Figure 5: **Communication clusters:** 4096 communication messages are embedded into low-dimensional representation using t-SNE [44] and is clustered using DBSCAN [11]. We visualize the images corresponding to the communication messages. We observed that the message clusters correspond to various meaningful phases throughout the task. The communication symbol of the purple cluster corresponds to when no doors are visible by either agent, and the light green cluster corresponds to when the red door is opened.

environment milestone events: for `FindGoal`, this is when the first agent reaches the goal; for `RedBlueDoors`, this is when the red door is opened. Since the identity of the agents that solve the task first does not matter, entropy plots are computed with respect to the *listener* agents (i.e., agents that receive vital information from others). In `FindGoal`, this corresponds to the last agent to reach the goal; in `RedBlueDoors`, this corresponds to the agent opening the blue door. For both environments, we see a sharp fall-off in entropy as soon as the first agents finish the task. In contrast, agents trained without autoencoding act randomly regardless of whether other agents have completed the task. This reaffirms that the agents trained with an autoencoder can effectively transmit information to other agents.

**Positive Signaling and Positive Listening**
We additionally investigate the two metrics suggested by [27] for measuring effectiveness of communication, *positive signaling* and *positive listening*. Since `ae-comm` agents have to communicate their learned representation, the presence of representation learning task loss means that `ae-comm` agents are intrinsically optimized for *positive signaling* (i.e., sending messages that are related to their observation or action). In Table 2, we report the increase in average reward when adding a communication channel, comparing `ae-comm` with its baselines; this metric is suggested by [27] to be a sufficient metric for *positive listening* (i.e., communication messages influence the behavior of agents). We observe that agent task performance improves most substantially in `ae-comm` with the addition of communication channel.

| methods | CIFAR | RedBlueDoors |
|---|---|---|
| | **gain** $\Delta r \uparrow$ | **gain** $\Delta r \uparrow$ |
| rl-comm | $0.017 \pm 0.022$ | $-0.030 \pm 0.155$ |
| rl-comm w/ bias [10] | $0.060 \pm 0.029$ | $0.552 \pm 0.155$ |
| ae-comm (ours) | $\mathbf{0.266 \pm 0.058}$ | $\mathbf{0.807 \pm 0.185}$ |

Table 2: **Performance gain with communication:** Positive listening as measured by the increase in reward after adding a communication channel. Performance of `ae-comm` agents improves more than the baselines.

## 6 Discussion and Societal Impacts

We present a framework for grounding multi-agent communication through autoencoding, a simple self-supervised representation learning task. Our method allows agents to learn non-differentiable communication in fully decentralized settings and does not impose constraints on input structures (e.g., state inputs or pixel inputs) or task nature (e.g., referential or non-referential). Our results demonstrate that agents trained with the proposed method achieve much better performance on a suite of coordination tasks compared to baselines.

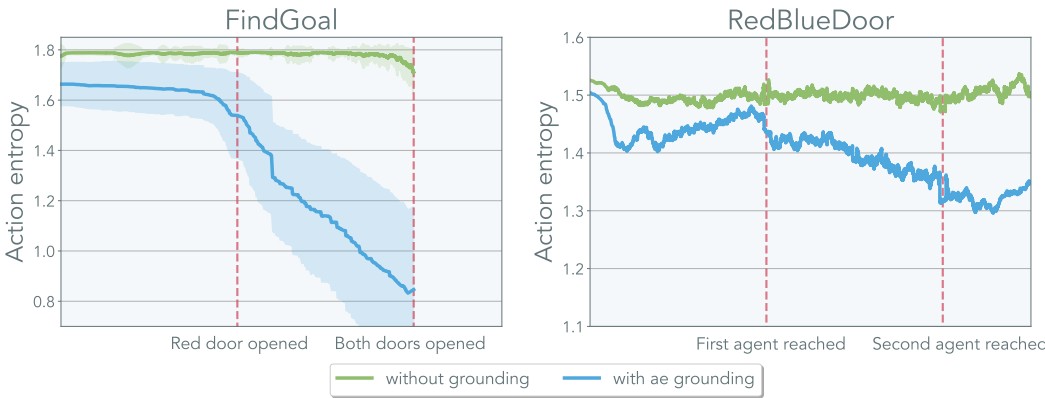

Figure 6: **Policy entropy with communication:** We visualize the entropy of the action policy throughout the task (lower is better). The graph is generated with 256 random episodes. For `FindGoal`, the entropy is measured on the last agent to enter the goal, and for `RedBlueDoors` the entropy is measured on the agent that opens the blue door (the second door). All 256 runs are aligned to the *dotted red lines* which corresponds to the time in which the first and second agent enters the goal for (`FindGoal`), and the time in which the red and the blue doors are opened for (`RedBlueDoors`). The model trained with an auto-encoder transmits messages that are effectively used by other agents.

We believe this work on multi-agent communication is of importance to our society for two reasons. First, it extends a computational framework under which scientific inquiries concerning language acquisition, language evolution, and social learning can be made. Second, unlike works in which agents can only learn latent representations of other agents through passive observations [45, 46], it opens up new ways for artificial learning agents to improve their coordination and cooperative skills, increasing their reliability and usability when deployed to real-world tasks and interacting with humans.

However, we highlight two constraints of our work. First, our method assumes that all agents have the same model architecture and the same autoencoding loss, while real-world applications may involve heterogeneous agents. Second, our method may fail in scenarios where agents need to be more selective of the information communicated since they are designed to communicate their observed information indiscriminately. We believe that testing the limits of these constraints, and relaxing them, will be important steps for future work.

We would also like to clarify that communication between the agents in our work is only part of what true "communication" should eventually entail. We position our work as one partial step toward true communication, in that our method provides a strong bias toward positive signaling and allows decentralized agents to coordinate their behavior toward common goals by passing messages amongst each other. It does not address other aspects of communication, such as joint optimization between a speaker and a listener. In the appendix section, we highlight the difficulty of training emergent communication via policy optimization alongside empirical results.

Another limitation of this work, which is also one concern we have regarding its potential negative societal impact, is that the environments we consider are fully cooperative. If the communication method we present in this work is to be deployed to the real world, we need to either make sure the environment is rid of adversaries, or conduct additional research to come up with robust counter-strategies in the face of adversaries, which could use better communication policies as a way to lie, spread misinformation, or maliciously manipulate other agents.

## Acknowledgements

We sincerely thank all the anonymous reviewers for their extensive discussions on and valuable contributions to this paper. We thank Lucy Chai and Xiang Fu for helpful comments on the manuscript. We thank Jakob Foerster for providing inspiring advice on multi-agent training. Additionally, TL would like to thank Sophie and Sofia; MH would like to thank Sally, Leo and Mila; PI would like to thank Moxie and Momo. This work was supported by a grant from Facebook.

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
