# OpenReview forum: "Learning to Ground Multi-Agent Communication with Autoencoders"
_NeurIPS.cc/2021/Conference — NeurIPS 2021 Poster_

### Official Review · Reviewer_tRBg · 2021-07-16

**Rating:** 6
**Confidence:** 4

**Summary:**

The paper proposes to learn the communication protocol in an emergent communication environment via an autoencoder; i.e. they train an autoencoder to learn a discrete representation of the image part of the observation, and use that as the message that gets broadcasted to all other agents at each timestep. The paper shows that this achieves significant gains over learning the communication protocol from scratch in a decentralized way.

**Limitations And Societal Impact:**

I think the limitations I mentioned above / framing could be improved, but otherwise no concerns.

**Main Review:**

This paper is very well written, with several beautiful plots and figures, many auxiliary experiments that help understand what's going on. It -looks- like a NeurIPS paper.

At the core of the paper is a simple experiment --- rather than learning a communication protocol from the environmental reward, have agents share a discrete representation of their state at each time step. It turns out that this works very well in the environments that the paper considers (a CIFAR-based referential game, and some GridWorld environments requiring communication). The reason for this is pretty intuitive: communicating a discrete representation of your state is really all of the information you need to solve these tasks (i.e. it's all the information that the other agents need). Decentralized communication protocols are hard to learn, so it makes sense that they perform way worse.

To me, this is an interesting experiment! I'm a bit surprised that nobody has thought of it before (to my knowledge), but autoencoding is definitely a strong communication prior, so it's a very reasonable thing to try.

That being said, I have a couple of concerns about the paper:
- I don't think the paper quite moves beyond "this is an interesting experiment" into "this could actually be useful / fundamental" territory. To do this, I would like to have seen either: (a) this approach compared to other methods, e.g. Jaques et al. or Eccles et al., showing that it outperforms them (I don't really like when a paper claims 'the effectiveness of previous approach X is limited, but don't compare to that approach); or (b) testing how the method works in environments where communicating only the state is not sufficient for coordination (e.g. communicating about past actions, or future intentions). I think the fact that doing RL on top of the autoencoder prior does *worse* is indicative that communicating about the state is indeed all you need for these tasks (or maybe that the RL component was poorly tuned).
- I'm not a big fan of the framing of the approach as 'grounding' the communication, implying that learning the communication protocol from scratch is 'ungrounded'. To me, if the agent learns to communicate with other agents in a grounded environment (such that it improves reward / is actually used), then that communication is grounded. The overall effect is that the paper seems to be a bit 'over-hyped'. This is reinforced by the allusion to cognitive science + evolutionary linguistics as motivation (which isn't wrong, per se, but feels a bit stretched).

That being said, compared to other papers in emergent communication I do think this paper comes close to meeting the bar (most other emergent communication papers don't compare to other papers in the field). It's still a nice paper. But I find myself wanting either more substance to the paper, or a modification in the framing.

All things considered, I'd rate this paper as borderline.

Small points:
- Reference missing on the straight-through estimator

**Time Spent Reviewing:**

2

---

> ### Author Response · Authors · 2021-08-10
> **Response to Reviewer tRBg**
>
> We thank the reviewer for the helpful feedback.
>
> **"more substance"**
>
> We agree that this is a fair concern. The closest to our work is Eccles et al. [7], so we implemented the Eccles et al. method (i.e. RIAL [13] with inductive biases) and ran additional experiments in the CIFAR Game environment and the RedBlueDoor environment. We tuned the target message entropy and the loss weights for the best performance of the Eccles et al. method, and report the results below. These results show that our method outperforms the Eccles et al. method by a margin. We hypothesize that the difference in performance results from our method imposing a much stronger prior for positive signaling.
>
>
> | average reward                      | CIFAR             | RedBlueDoors      |
> |-------------------------------------|-------------------|-------------------|
> | no-comm                             | 0.082 ± 0.009     | 0.123 ± 0.096     |
> | rl-comm                             | 0.099 ± 0.013     | 0.174 ± 0.009     |
> | rl-comm-with-biases (Eccles et al.) | 0.142 ± 0.019     | 0.729 ± 0.072     |
> | ae-comm (ours)                      | **0.348 ± 0.041** | **0.984 ± 0.002** |
>
> (All confidence intervals reported are 95% confidence intervals, as in Eccles et al.)
>
> We will continue to run experiments on the FindGoal environment, since those take longer time. We will update our response once the remaining experiments are completed, and will include all these results in the revised version of our paper.
>
> In addition, we have added a baseline where the speaker transmits a latent state from the policy network without autoencoding. Below, we report additional comparisons between this newly added baseline ("fc-comm") and our method ("ae-comm") on all three environments. With these results, we further confirm the hypothesis that autoencoding is indeed a necessary component for our method.
>
> | average reward | CIFAR             | RedBlueDoors      |
> |----------------|-------------------|-------------------|
> | fc-comm        | 0.107 ± 0.021     | 0.382 ± 0.022     |
> | ae-comm (ours) | **0.348 ± 0.041** | **0.984 ± 0.002** |
>
> | average episode length | FindGoal         |
> |------------------------|------------------|
> | fc-comm                | 188.3 ± 33.6     |
> | ae-comm (ours)         | **103.5 ± 20.2** |
>
> We also want to note that our "no comm" baseline is in fact an A3C version of RIAL [13].
>
> **"a modification in the framing"**
>
>
> Thank you for pointing out this issue with framing, which we agree with. We used the term "grounding" to emphasize that our work adds a stronger prior for agents to associate their communication with observations. We have not yet found a better term, but will definitely tone down the allusions to cognitive science and evolutionary linguistics in the introduction. (Those allusions were insisted on by our advisor :) .)
>
> **Small points**
>
> Thank you for the suggestions. We'll fix this in the next version of our paper.

---

> > ### Author Response · Authors · 2021-08-17
> > **Remaining results for Eccles et al.**
> >
> > Here are the remaining results for Eccles et al.:
> >
> >
> > | average episode length | FindGoal         |
> > |------------------------|------------------|
> > | no-comm                | 169.0 ± 26.8     |
> > | rl-comm                | 184.8 ± 25.2     |
> > | rl-comm-with-biases (Eccles et al.) | 157.99 ± 12.3     |
> > | ae-comm (ours)         | **103.5 ± 20.2** |
> >
> >
> > We will include the full results in our revised paper.

---

> > > ### Comment · Reviewer_tRBg · 2021-08-29
> > > **Response**
> > >
> > > I thank the authors for their reply. I appreciate the addition of the baselines (particularly the fc-comm and latent-comm mentioned in a response to another reviewer). I'm now lean towards acceptance, though my score remains a 6.

---

### Official Review · Reviewer_rwT3 · 2021-07-16

**Rating:** 6
**Confidence:** 4

**Summary:**

This paper proposes an approach towards encouraging the development of communication protocols in multi-agent reinforcement learning systems. The authors rightly observe that naive training of a communication policy according to task reward often fails, due to high variability in the initial messages, the difficulty of connecting this communication to reward, and lack of common grounding among the agents.

The authors' idea is to imbue agents with a message-generating mechanism cast as an autoencoder, where the agents aim to generate discrete messages that can reconstruct the observations of the agents. Thus, the message generated for a given timestep is essentially a discretized summary of an agents' observation at that time. This encourages the messages to be maximally informative about an agents' current state—something that is shown by qualitatively.

The authors evaluate this communication idea on a variety of environments, from a reference game variant to a few multi-agent minigrid environments, and show that their autoencoded communication outperforms (1) non-communicative baselines and (2) communication policies trained with standard RL objectives. They then have a few analyses that show that the agents are making use of the generated messages, in that (1) messages are informative about agents' current states (by design) and (2) messages affect other agents' entropy over decisions once they indicate a change in state.

**Limitations And Societal Impact:**

Yes

**Main Review:**

Overall, this is an interesting and clear paper with a nice idea at its core, but I see an obvious missing comparison to related work and some more philosophical issues with the papers' framing that make me hesitant to recommend the paper outright. I'm looking forward to the other reviews and the author response.

## Strengths

- The paper is well written and clear.
- It has several clean experiments across a variety of environments that clearly show the benefits of autoencoded communication over non-communication and naive baselines.
- The idea of messages as autoencoding agent observations is an interesting one—it has appeared here and there in the literature (see missing references) but is clearly presented here and will be of interest to those working in MARL. It's likely that such an objective can be combined with more directly communicative objectives, to both encourage grounding and optimize for communication.
- While some more analyses could be done, I buy the authors' analysis in Section 5.5 that the agents learn to use and understand the messages, both due to the inherent *positive signaling* bias of the autoencoding objective and Figure 7 (which neatly shows that agents become more confident when their partner reaches a subgoal of the environment, and that this confidence can only be a result of the availability of the partners' message).

## Weaknesses

I have two major concerns with this work, and some other issues/questions.

### No comparison to Eccles et al.

Eccles et al. [7] is perhaps the most relevant paper for this work. It too suggests that naively optimizing communication via the typical MARL reward signal is brittle and often fails, and propose several auxiliary objectives. Yet the paper is quickly dismissed in L81-83 as "requiring task-specific hyperparameter tuning" and "limited effectiveness." Task-specific hyperparameter tuning is a fair criticism, and one claimed advantage of the current method could be its relative ease-of-implementation. But "limited effectiveness" is an unsubstantiated claim, and taken together these reasons are not sufficient to completely discard comparison to their proposed approaches. Yet no such comparisons are done in this paper.

### Is this "communication"?

My other concern is more conceptual: it makes me uncomfortable to imagine the experiments in this paper as true "communication" among agents, since the learned messages are not trained via any inherent communicative objective (i.e.: generate some message to maximize some reward given some dialog context). Rather, the messages are generated solely from an autoencoding objective: essentially learning a discrete representation of an agents' current observation space (without any regard to what influence this has on the ultimate success of the group).

One question I have for the authors is, is this a significantly different setup than simply doing centralized MARL, where each agent has full access to the other agents' observation space and internal state? Especially if the agents policies have some discrete latent variable representation which we could then label as "messages"?

In some ways this paper is an extension of the Eccles et al. paper: one part of it proposes a bias for positive signaling (that we generate messages correlated with our current state), and this paper takes it to an extreme where we actually don't consider *anything* besides positive signaling (i.e. the *only* objective is to maximize MI between messages and observations).

This means that many of the purported benefits of the "communication" learned by the agents in this paper seem given "for free". For example, the authors admit in L282 that the autoencoder agents are intrinsically optimized for "positive signaling" - but indeed, the messages are generated with no other objective besides positive signaling. Similarly, the visualization in Figure 5 suggests that "messages" cluster according to episode state. But this is unsurprising: we would expect an optimal policy's internal representations to also cluster according to episode state.

I can see this somewhat philosophical observation resulting in more concrete limitations of the proposed method: if the communication channel is very limited compared to the agents observation state—either because of limited bandwidth (i.e. vocabulary) or a cost to sending messages—then such an autoencoding objective may not work. (Indeed we need more details about the communication channel size, as I'll mention again below). In general, there might be many tasks when agents must learn to be more selective with the information communicated at each timestep, and a purely "descriptive" objective like the one presented here might fail, and we need agents that learn *what* to communicate. (This is the same reason why we humans do not constantly vocalize our environmental state every passing second). Admittedly, these constraints are not often considered in current MARL research, and more communicative objectives could be combined with the proposed autoencoding objective (i.e. a "first step towards grounding" as in L58-59, though again, this idea is similar to Eccles et al.).

### Other issues/questions

- Insufficient description of baselines in L223-224, specifically with regards to learning the communication via a separate policy network. Why is the communication module in the listener module instead of the speaker module (L224)? isn't the autoencoded representation of the speaker model precisely some sort of discrete message - here you are learning a policy network on top of this to go from this message to another message?
- L240: I don't understand how figure 3c shows that ae-comm agents are able to complete the episode much faster than other agents. All 3 lines seem to be within the shaded error regions? What does the error region represent? In cases that are visually ambiguous like this one, it would be good to have some sort of significance test.
- I'm very suspicious of the RL-communication policy not working above chance (no communication) in something as simple as the CIFAR referential game. This is an odd conclusion given the large literature on emergent communication in referential games that finds that we can learn successful (albeit uninterpretable) communication protocols with policy gradient, differentiable relaxations, mixtures of both, etc. Could the authors explain this result more?
- Would be nice to explicit calculate the proposed CIC metric (i.e. mutual information between messages and other agents' actions) in L274-276, which is precisely what Lowe et al. [22] suggests.
- It's not clear to me how Figure 6 illustrates positive listening. Autoencoder loss and task performance are correlated, but that's because we're jointly optimizing the two - it doesn't mean that the increase in task performance is *caused* by the improved autoencoder reconstruction. (I do buy the Figure 7 analysis more).

## Missing references

The idea generating messages that the model itself can interpret (in this case: reconstruct its own state) is similar to the obverter communication method proposed by Choi et al., Compositional Obverter Communication Learning From Raw Visual Input, ICLR 2018

## Minor

- L146 Missing Citation
- The communication spaces C could be more precisely defined across both environments.
- Figure 6 would be more clear if Loss was labeled "Autoencoder Loss"

**Time Spent Reviewing:**

3

---

> ### Author Response · Authors · 2021-08-10
> **Response to Reviewer rwT3 (1/2)**
>
> We thank the reviewer for this constructive feedback.
>
> **No comparison to Eccles et al.**
>
> Thank you for this suggestion. We ran additional experiments to compare our approach with that of Eccles et al. (i.e. RIAL [13] with inductive biases) in the CIFAR Game environment and the RedBlueDoor environment. We tuned the target message entropy and the loss weights for the best performance of the Eccles et al. method, and report the results below. These results show that our method outperforms the Eccles et al. method by a margin and is more stable. We hypothesize that the difference in performance results from our method imposing a much stronger prior for positive signaling.
>
> | average reward                      | CIFAR             | RedBlueDoors      |
> |-------------------------------------|-------------------|-------------------|
> | no-comm                             | 0.082 ± 0.009     | 0.123 ± 0.096     |
> | rl-comm                             | 0.099 ± 0.013     | 0.174 ± 0.009     |
> | rl-comm-with-biases (Eccles et al.) | 0.142 ± 0.019     | 0.729 ± 0.072     |
> | ae-comm (ours)                      | **0.348 ± 0.041** | **0.984 ± 0.002** |
>
>
> (All confidence intervals reported are 95% confidence intervals, as in Eccles et al.)
>
> We will continue to run experiments on the FindGoal environment, since those take longer time. We will update our response once the remaining experiments are completed, and will include all these results in the revised version of our paper.
>
> In addition, we have added a baseline where the speaker transmits a latent state from the policy network without autoencoding. Below, we report additional comparisons between this newly added baseline ("fc-comm") and our method ("ae-comm") on all three environments. With these results, we further confirm the hypothesis that autoencoding is indeed a necessary component for our method.
>
>
> | average reward | CIFAR             | RedBlueDoors      |
> |----------------|-------------------|-------------------|
> | fc-comm        | 0.107 ± 0.021     | 0.382 ± 0.022     |
> | ae-comm (ours) | **0.348 ± 0.041** | **0.984 ± 0.002** |
>
> | average episode length | FindGoal         |
> |------------------------|------------------|
> | fc-comm                | 188.3 ± 33.6     |
> | ae-comm (ours)         | **103.5 ± 20.2** |
>
>
>
> **Is this "communication"?**
>
> We agree that the communication between the agents in our work is only part of what true "communication" should eventually entail. As mentioned in our introduction, the way we position our work is that it solves some of the same problems that true communication is meant to solve: it allows decentralized agents to coordinate their behavior toward common goals by passing messages amongst each other. In the revision, we will clarify exactly what we mean by "communication", and situate our work as one partial step toward true communication -- we will explain that our method provides a strong bias toward positive signaling but does not address other aspects of communication, such as joint optimization between a speaker and a listener.
>
>
> **Is this a significantly different setup than simply doing centralized MARL, where each agent has full access to the other agents' observation space and internal state?**
>
> Our method is *not* equivalent to giving full access to other agents’ observations and internal states. We agree that it would be a stretch to call that communication since the bandwidth of the communication channel would have to be quite large to carry all that information. Rather, our method gives access to an autoencoded *representation* of the other agents’ observations, and this representation takes a form that could be communicated as a compact message. We have also added an additional experiment relevant to this point: rather than broadcasting autoencoded observations, we broadcast a latent state from the policy network without autoencoding. We find that the latter performs much worse (see the comparison between "ae-comm" and "fc-comm" above). This demonstrates that just sharing an internal state variable is not enough, it is important to share the right state representation, and autoencoding is one surprisingly effective choice.
>
> **More concrete limitations of the method**
>
> We truly appreciate your careful reading of our paper, and agree with what you have pointed out: that our work in some way takes the positive signaling bias to an extreme, and that this leads to potential failure of the method in scenarios where agents need to be more selective of the information communicated. We will include these limitations in the discussion section of our revised paper.
>
> We think that addressing these more challenging communication problems is indeed a very important direction to explore. In fact, we have tested our method on environments with much more limited communication channels, and naive application of our methods would not achieve much better results than baselines. Optimizing RL (or other more communicative objective) on top of the autoencoding prior involves non-trivial work. We will include discussion of these failures in our revised paper.
>
> However, we believe that these are not fundamental limitations of our work -- our method already makes a novel contribution achieving successful decentralized and non-differentiable multi-agent communication in our current set of environments. In the future, we hope to make more advances towards solving these harder communication problems, by building on this work.
>
> **Differences between our work and Eccles et al.**
>
> We also would like to emphasize that there are still substantial differences between our method and the Eccles et al. method, besides the difference in performance. The core approach of Eccles is to add biases on top of RIAL [13], while ours is to separate *signaling* and *listening* into completely different learning stages. These are two very different ideas conceptually.
>
> **Poor performance of RL-communication policy**
>
> > "I'm very suspicious of the RL-communication policy not working above chance in something as simple as the CIFAR referential game"
>
> In fact, at the beginning of this project, we had the exact same doubt: how can RL communication policy perform this poorly? However, as we dived deeper into the emergent communication, we found that this result is actually not that surprising. One thing to note is that our rl-comm baseline can be seen as an A3C version of the RIAL [13] method. And in the original paper [13], performance of RIAL was very close to that of a no-communication baseline on the multi-step MNIST Games, a task similar to CIFAR Games in our paper. Our hypothesis is that optimization is harder whenever the joint-exploration problem for learning speaker and listener policies is introduced.
>
> As mentioned in our related works section, learning to communicate with decentralized training and non-differentiable communication is a very challenging task that hasn't been satisfactorily solved. Despite the large literature, we found that most cases where communication successfully emerged rely on differentiable communication. These findings from literature review were exactly what motivated us to propose our method.
>
> **Ambiguity of figure 3c**
>
> We realize that there are three factors that may have contributed to the visual ambiguity in Figure 3c (i.e. large overlap of error regions):
> - The small number of runs we had (over 5 random seeds).
> - The small number of trials for each evaluation data sample (over 10 trials).
> - We reported the standard deviation of the moving average in each graph, rather than a 95% confidence interval as the shaded region (as in Eccles et al.).
>
> It is also worth noting that the FindGoal task can be solved without communication, unlike the other two tasks. The difference in agent performance is larger when there are more agents and/or there is a larger map. Given our limited computational resources and time constraints for the rebuttal, we found with our current settings to achieve the best balance between valid results and training time.
>
> During the past few days, we have been running additional experiments (5 more seeds) on the FindGoal environment, and evaluating on 100 trials per data point, to lessen this ambiguity. We have re-calculated the statistics and are reporting below the mean and 95% confidence intervals. We hope these demonstrate the stability and effectiveness of our approach.
>
> | average episode length | FindGoal         |
> |------------------------|------------------|
> | no-comm                | 169.0 ± 26.8     |
> | rl-comm                | 184.8 ± 25.2     |
> | ae-comm (ours)         | **103.5 ± 20.2** |
>
> We will continue to run more experiments and incorporate the results in the next revision. We will do the same for other environments as well, so that experiment results in our paper are standardized.

---

> > ### Author Response · Authors · 2021-08-10
> > **Response to Reviewer rwT3 (2/2)**
> >
> > **Positive listening and CIC metric**
> >
> > > "It's not clear to me how Figure 6 illustrates positive listening"
> >
> > Thank you for pointing out the lack of scientific rigor here. We agree that Figure 6 does not sufficiently illustrate positive listening and will rework this section in the revised paper.
> >
> > Specifically, we will move Figure 6 to appendix and instead clarify the positive listening aspect with a table reporting increase in reward when adding a communication channel. This is suggested by [22] to be a sufficient metric for positive listening.
> >
> > | Δr                                  | CIFAR             | RedBlueDoors      |
> > |-------------------------------------|-------------------|-------------------|
> > | rl-comm                             | 0.017 ± 0.022     | -0.030 ± 0.155    |
> > | rl-comm-with-biases (Eccles et al.) | 0.060 ± 0.029     | 0.552 ± 0.155     |
> > | ae-comm (ours)                      | **0.266 ± 0.058** | **0.807 ± 0.185** |
> >
> >
> > > "Would be nice to explicit calculate the proposed CIC metric"
> >
> > We have actually considered using the CIC metric proposed in [22], but found implementation to be infeasible in our case because our communication policy is not parameterized by a probability distribution like an RL communication policy. Furthermore, even the one-step CIC requires iteration through counterfactuals of all agents and all possible communication messages; computational cost of doing so scales exponentially with the number of agents and communication space, and we found our CIC metric calculation on the rl-comm baseline to be very slow.
> >
> > **Insufficient description of baselines**
> >
> > For the rl-comm baseline, we do not make a distinction between the listener module and the speaker module. The communication policy is produced in a way similar to the environment policy; there is no autoencoding involved. In fact, our rl-comm baseline can be seen as an A3C version of RIAL [13] without parameter sharing.
> >
> > We hope this clarifies, and will make sure to describe the baselines more clearly in our revised paper.
> >
> > **Missing references**
> >
> > Thank you for pointing this out. Our work and Choi et al. indeed share a similar high-level idea. We will cite Choi et al. in our revised paper and discuss this similarity.
> >
> > We would also like to emphasize that the two works are very different in terms of specific type of communication, model architecture, and focus of the paper. Most importantly, Choi et al. uses a fully differentiable model architecture for learning a communication protocol, while the communication in our work is non-differentiable.
> >
> > **Minor comments**
> >
> > Thank you very much for these comments. We'll remedy them in the next version of our paper.

---

> > > ### Comment · Reviewer_rwT3 · 2021-08-10
> > > **Response**
> > >
> > > Thanks for the very detailed response to my review! Could authors clarify what fc-comm looks like more? "Latent state from the policy network without autoencoding" - is this a discrete state that looks the same as the autoencoded messages?

---

> > > > ### Author Response · Authors · 2021-08-10
> > > > **Clarification for "fc-comm"**
> > > >
> > > > Yes, in "fc-comm", the speaker transmits discrete states that look the same as the autoencoded messages in "ae-comm".
> > > >
> > > >
> > > > These discrete states come from the policy network without autoencoding. For fair comparison, a fully connected layer is added on top of the policy network states, such that the learned messages for "fc-comm" are discretized and have the same shape and range as messages in "ae-comm".

---

> > > > > ### Comment · Reviewer_rwT3 · 2021-08-10
> > > > > **Response**
> > > > >
> > > > > Thanks for the clarification!
> > > > >
> > > > > I greatly appreciate the inclusion of the Eccles et al. results which strengthen the paper.
> > > > >
> > > > > **Re: RL-comm on CIFAR**
> > > > >
> > > > > [Lazaridou et al. 2017](https://arxiv.org/abs/1612.07182) explores a similar communication task which also depends on non-differentiable communication and RIAL-style training. Could authors clarify the differences that result in successful communication in Lazaridou et al., but unsuccessful communication in their setting? One reason why RIAL doesn't work well in the original paper is because this involves a *multi-step* MNIST game—note that RIAL does indeed improve performance for the simpler color-digit game.
> > > > >
> > > > > I have read the other reviews and responses, and given the partial comparisons to Eccles et al. I will raise my score to 6. if the above point of confusion is resolved and full results for Eccles et al. are obtained, I would consider raising my score further.

---

> > > > > > ### Author Response · Authors · 2021-08-12
> > > > > > **Follow-up response to Reviewer rwT3**
> > > > > >
> > > > > > Thank you for the positive feedback.
> > > > > >
> > > > > > > Lazaridou et al. 2017 explores a similar communication task which also depends on non-differentiable communication and RIAL-style training. Could authors clarify the differences that result in successful communication in Lazaridou et al., but unsuccessful communication in their setting?
> > > > > >
> > > > > > Regarding Lazaridou et al. 2017, we want to highlight that their task is easier than our CIFAR Game and MNIST Game in [13], and their training is not RIAL because their agents do not interact through communication. Specifically, we identify the following differences between our work and their work:
> > > > > >
> > > > > > - Lazaridou et al. explores a task in which 2 agents use RL to predict which image from a target-distractor pair is the actual target. These 2 agents are specialized to be either a speaker or a listener, but not both. This means that the sender and receiver agents in their setting do not actually "interact" via communication; the sender only learns to produce a message, and the receiver only learns to decode the message. Hence, the problem boils down to learning to auto-encode by reinforcement learning. On a high-level, a pair of sender-receiver agents in their setup is closer to a single agent in our setup, where the "sender" can be seen as the speaker module and "receiver" the listener module.
> > > > > >
> > > > > > - Moreover, in the setup of Lazaridou et al., "images" given to the agents are not raw pixels from the original images, but their latent representations obtained from a VGG pretrained on ImageNet. Since these representations already have a strong prior on natural images, using them as inputs allows agents to sidestep the need of learning to classify the images. This greatly eases the task.
> > > > > >
> > > > > > - Their model architecture further simplifies the task by hard-coding biases towards successful binary classification. Upon receiving a message transmitted by the sender, the receiver explicitly computes dot similarity scores between this message and the target/distractor image embeddings. The receiver then uses these similarity scores to initialize the final action distribution. This implicitly forces the transmitted message to contain relevant information that could be used to distinguish the target image and the distractor image, easing the exploration problem in communication.
> > > > > >
> > > > > > Based on these differences, we find it unsurprising that communication indeed works in their setting, and the challenges in our "rl-comm" setup remains.
> > > > > >
> > > > > >
> > > > > > > full results for Eccles et al.
> > > > > >
> > > > > >
> > > > > > Regarding the remaining results for Eccle et al., we are still working on the experiments for FindGoal environment and will update them once we have the numbers.

---

> > > > > > > ### Author Response · Authors · 2021-08-17
> > > > > > > **Remaining results for Eccles et al.**
> > > > > > >
> > > > > > > Here are the remaining results for Eccles et al.:
> > > > > > >
> > > > > > >
> > > > > > > | average episode length | FindGoal         |
> > > > > > > |------------------------|------------------|
> > > > > > > | no-comm                | 169.0 ± 26.8     |
> > > > > > > | rl-comm                | 184.8 ± 25.2     |
> > > > > > > | rl-comm-with-biases (Eccles et al.) | 157.99 ± 12.3     |
> > > > > > > | ae-comm (ours)         | **103.5 ± 20.2** |
> > > > > > >
> > > > > > >
> > > > > > > We will include the full results in our revised paper.

---

### Official Review · Reviewer_HBSZ · 2021-07-17

**Rating:** 7
**Confidence:** 3

**Summary:**

The goal of the paper is to show that learning a representation to ground language can improve communication in multi-agent settings.

Their method: each agent encodes an image and the other agents messages (through a small conn-net and mlp, respectively). A GRU-based policy outputs an action. A communication autoencoder predicts a message from the image feature and is trained with an auxiliary construction objective (i.e., to reconstruct the image).

They evaluate their model on a referential game with CIFAR images and two grid environments: one where the agents must reach a goal collaboratively and another where one agent has to open a door for another to pass.

They show that their method with an auto encoder strongly outperforms the non-autoencoder variants on the CIFAR task and door task.

**Ethical Concerns:**

There are no ethical concerns.

**Limitations And Societal Impact:**

The authors discuss societal impacts.

**Main Review:**

Originality: their main contribution is the design of a multi-agent communication architecture that uses an auto encoder as an auxiliary loss.

Quality: The work is evaluated on a very limited set of tasks and there are no baselines they compare against. The main result (Figure 3) is an ablation. The experiments were also conducted without any hyper parameter tuning.

Clarity: The introduction is very high level, which makes it difficult to understandl. I think it would help to ground the introduction in more specific technical contributions in related work to make it more accessible.

Significance: It's difficult to contextualize the significance of the work because of it's lack of comparisons to other multi-agent work.

Minor comments:
- The visualization on the right side of figure 1 is confusing. Specifically, why are there bidirectional arrows between the communication encoder and message (and policy network and action) if it's generating the message? It may help to visualize these as modules instead, like the encoders below.

Here are some other works on multi-agent interaction and communication that may be relevant:
Learning Latent Representations to Influence Multi-Agent Interaction. Xie et. al., 2020.
TarMAC: Targeted Multi-Agent Communication. Das et. al., 2019.
Emergent Linguistic Phenomena in Multi-Agent Communication Games. Graesser et. al., 2019.

**Time Spent Reviewing:**

2

---

> ### Author Response · Authors · 2021-08-10
> **Response to Reviewer HBSZ**
>
> We thank the reviewer for this valuable feedback.
>
> **"limited set of tasks"**
>
> In our work, we have included a variety of tasks that cover scenarios that are (1) referential or non-referential, (2) ordinal or non-ordinal, and (3) two-agent or generalized multi-agent. Note that there has not been any standardized benchmark task in the emergent communication literature, and the variety and complexity of our tasks are comparable to tasks in other works in the field. As Reviewer rwT3 acknowledged, our work has "a variety of environments, from a reference game variant to a few multi-agent minigrid environments".
>
> **"no baselines" / "lack of comparison to other multi-agent work"**
>
> In fact, we have compared our method with three baselines (described in section 5.2), and Figure 3 shows a comparison between our method and two of the baselines ("no comm" and "rl comm"). We would like to clarify again that the "rl-comm" is equivalent to an A3C version of RIAL [13], which Reviewer ux54 had also mentioned. And the "no comm" baseline is a baseline commonly used across the emergent communication literature; it is shown in [22] that comparison with this baseline "provides a strong indicator that communication is present".
>
> We have also added comparisons to two other baselines. We describe the details below.
>
> 1. "fc-comm", a variant of our method where the speaker transmits a latent state from the policy network without autoencoding
>
> Here, we report additional comparisons between "fc-comm" and our method ("ae-comm") on all three environments. With these results, we further confirm the hypothesis that autoencoding is indeed a necessary component for our method.
>
>
> | average reward | CIFAR             | RedBlueDoors      |
> |----------------|-------------------|-------------------|
> | fc-comm        | 0.107 ± 0.021     | 0.382 ± 0.022     |
> | ae-comm (ours) | **0.348 ± 0.041** | **0.984 ± 0.002** |
>
> | average episode length | FindGoal         |
> |------------------------|------------------|
> | fc-comm                | 188.3 ± 33.6     |
> | ae-comm (ours)         | **103.5 ± 20.2** |
>
>
> 2. Eccles et al. [7] (i.e. RIAL [13] with inductive biases), a state-of-the-art method from the literature
>
> As suggested by Reviewer rwT3, Reviewer tRBg, and Reviewer ux54, we further compare our work against Eccles et al. [7], which is RIAL [13] with additional inductive biases for positive listening and positive signaling. We tuned the target message entropy and the loss weights for the best performance of the Eccles et al. method.
>
> Here, we report results in the CIFAR Game environment and the RedBlueDoor environment. These results show that our method outperforms the Eccles et al. method by a margin and is more stable.
>
> | average reward                      | CIFAR             | RedBlueDoors      |
> |-------------------------------------|-------------------|-------------------|
> | no-comm                             | 0.082 ± 0.009     | 0.123 ± 0.096     |
> | rl-comm                             | 0.099 ± 0.013     | 0.174 ± 0.009     |
> | rl-comm-with-biases (Eccles et al.) | 0.142 ± 0.019     | 0.729 ± 0.072     |
> | ae-comm (ours)                      | **0.348 ± 0.041** | **0.984 ± 0.002** |
>
>
> (All confidence intervals reported are 95% confidence intervals, as in Eccles et al.)
>
> We will continue to run experiments on the FindGoal environment, since those take longer time. We will update our response once the remaining experiments are completed, and will include all these results in the revised version of our paper.
>
> **"experiments were conducted without any hyperparameter tuning"**
>
> This is an incorrect statement. Our results are reported after thorough hyperparameter tuning. We will make this clear and include the details of our hyperparameter sweeps in the revised appendix.
>
> **"some other works on multi-agent interaction and communication that may be relevant"**
>
> Thank you for the suggested works. We have carefully read through these works, and will include them in our next revision. Below, we highlight the differences between our work and these works.
> - Xie et al. focuses on relieving the scalability issue of opponent modeling by capturing latent strategies of other agents.
> - Das et al. uses centralized training and decentralized execution (CTDE), while one goal for our paper is to learn multi-agent communication without CTDE.
> - Graesser et al. relies on supervised learning for training their agents’ communication. Furthermore, their focus is on the emergent linguistic behaviors that are also observed in natural language.
>
> **Minor comments**
>
> We will revise our figure to make the visualization clearer.

---

> > ### Author Response · Authors · 2021-08-17
> > **Remaining results for Eccles et al.**
> >
> > Here are the remaining results for Eccles et al.:
> >
> >
> > | average episode length | FindGoal         |
> > |------------------------|------------------|
> > | no-comm                | 169.0 ± 26.8     |
> > | rl-comm                | 184.8 ± 25.2     |
> > | rl-comm-with-biases (Eccles et al.) | 157.99 ± 12.3     |
> > | ae-comm (ours)         | **103.5 ± 20.2** |
> >
> >
> > We will include the full results in our revised paper.

---

> > > ### Comment · Reviewer_HBSZ · 2021-08-22
> > > **Response**
> > >
> > > Thank you for your detailed follow up. It would be helpful to state explicitly in the paper the equivalence between rl-comm and [13], as you have directly clarified here.
> > >
> > > The extended baselines and clarifications provided have alleviated many of my concerns, and I raised my score accordingly. They will make a strong addition to the final version of the paper.

---

### Official Review · Reviewer_ux54 · 2021-08-04

**Rating:** 7
**Confidence:** 4

**Summary:**

In this paper, the authors present a novel method for learning decentralized communication policies in fully cooperative multi-agent RL. The method is based on autoencoding agent observations and using these as messages, thus sidestepping the usual "chicken and egg" problem for simultaneously learning speaker and listener policies, which is a hard joint-exploration problem. The authors present results in which this method outperforms both non-communication and RIAL baselines, and include some analyses and ablations that shed light on how and why this method is successful.

**Limitations And Societal Impact:**

The limitations of the work should be discussed more clearly. For example, is it essential that agents have identical model architectures, as stated at the bottom of page 3? If so, this seems like quite a strong limitation, since most real-world applications may involve heterogeneous agents. On the other hand, if this limitation can be reduced or mitigated, the authors should provide experiments to this effect, or strong arguments for future work.

The societal impact is well addressed on page 9.

**Main Review:**

Positive aspects

- The paper establishes a convincing argument and method for grounding decentralized communication via autoencoding.
- The introduction and abstract are well-written, and most of the relevant literature is cited clearly.
- The method is elegant and well-described, and in my opinion the results would therefore be reproducible.
- The proposed method outperforms RIAL and no communication baselines on several different fully cooperative tasks.
- The t-SNE analysis in Figure 5 and the entropy analysis in Figure 7 are convincing methods that uncover the mechanisms underlying improved performance of agents under this method.

Aspects requiring improvement

- The paper seems to rely on a strong assumption that all agents have the same model architecture, and the same autoencoding loss. This reduces the impact of the work. It would be very interesting to have further experiments that assess to what extent this constraint is required. See also comments on "limitations" below.
- An important baseline is missing. Since the "speaker" is transmitting exactly an autoencoded latent state, a good baseline would be one where the speaker transmits a latent state from the policy network, without autoencoding. This would help to establish that it is the autoencoding itself that is necessary, rather than merely the simplification in the multi-agent learning dynamics that comes from fixing the speaker policy to be a latent state.
- The authors invent their own environments rather than using environments from the literature. This means that their work cannot be quantitatively compared to previous papers like (in their citations) [7,13,16]. Could the authors add one additional environment from the existing literature to make this comparison easier? In particular, does their method outperform some of the methods with inductive biases? To what extent does their method regain the "topline" performance afforded by fully differentiable centralized models like DIAL?
- The results in Figure 3(c) are inconclusive. The authors claim that AE comm is better, but the confidence intervals on their plot suggest that no such conclusion can be drawn. Could the authors comment on the statistically validity of their claim? If the task is simply too easy to see a meaningful difference between the 3 methods, perhaps it would be better to try a harder task here?
- Parts of the Experiments section are unconvincing. In particular, I did not find the claims about positive listening and positive signaling sufficiently rigorous: it feels like the authors are observing a correlation rather than demonstrating a causal relationship. This section would benefit from reworking or removing. Moreover Section 5.4 misses important hypotheses about why adding RL to the AE loss doesn't help. This might be because the optimization problem is harder when you reintroduce the "chicken and egg" joint-exploration problem for learning speaker and listener policies at the same time. Alternatively, or additionally, it might be because the multi-objective optimization problem is harder than the single-objective optimization problem, so training requires more iterations. It would be useful to have further experiments here to disambiguate these possibilities with the representation-learning hypothesis which the authors present.
- Some relevant citations are missing. For example there has been much recent work on learning latent representations to influence behavior in multi-agent systems, e.g. Xie, Losey, Tolsma, Finn and Sadigh 2020; Zintgraf, Devlin, Ciosek, Whiteson, Hofmann 2021. It would be useful to contextualise this contribution with reference to those papers.

Small comments

- In several places the authors refer to "cooperative" games where they could more precisely say "fully cooperative" or "common payoff" to distinguish from mixed-motive games, which are sometimes also referred to as cooperative.
- In lines 233 and 237 the names of the environments are the wrong way round, I believe.
- In line 214, the authors could consider using the more precise definition of "cheap talk" from Farrer 1987.
- The authors might want to consider citing Dafoe et. al, "Cooperative AI", 2020 in their introduction; this review contains a section on the importance of grounding.
- In their introduction, the authors may wish to also discuss the importance of the gestural modality in the origins of human communication, as in Tomasello, "Origins of Human Communication", 2008.

**Time Spent Reviewing:**

2

---

> ### Author Response · Authors · 2021-08-10
> **Response to Reviewer ux54**
>
> We thank the reviewer for this helpful feedback.
>
> **Missing baseline**
>
> We have added a baseline where the speaker transmits a latent state from the policy network without autoencoding. Below, we report additional comparisons between this newly added baseline ("fc-comm") and our method ("ae-comm") on all three environments. With these results, we further confirm the hypothesis that autoencoding is indeed a necessary component for our method.
>
>
> | average reward | CIFAR             | RedBlueDoors      |
> |----------------|-------------------|-------------------|
> | fc-comm        | 0.107 ± 0.021     | 0.382 ± 0.022     |
> | ae-comm (ours) | **0.348 ± 0.041** | **0.984 ± 0.002** |
>
> | average episode length | FindGoal         |
> |------------------------|------------------|
> | fc-comm                | 188.3 ± 33.6     |
> | ae-comm (ours)         | **103.5 ± 20.2** |
>
>
>
> **Comparison to previous works**
>
> We thought about re-implementing both the environments and algorithms from existing literature, but this is non-trivial work given the time constraint. Also, there is no standard environment in the emergent communication literature. We want to note that our CIFAR Game environment is similar to the MNIST Game environment in [13], and the no-comm baseline achieves similar performance on CIFAR Game as the RIAL on MNIST Game.
>
> For easier comparison, we have run additional experiments to compare our approach with that of Eccles et al. (i.e. RIAL [13] with inductive biases) in the CIFAR Game environment and the RedBlueDoor environment. We tuned the target message entropy and the loss weights for the best performance of the Eccles et al. method, and report the results below. These results show that our method outperforms the Eccles et al. method by a margin and is more stable. We hypothesize that the difference in performance results from our method imposing a much stronger prior for positive signaling.
>
> | average reward                      | CIFAR             | RedBlueDoors      |
> |-------------------------------------|-------------------|-------------------|
> | no-comm                             | 0.082 ± 0.009     | 0.123 ± 0.096     |
> | rl-comm                             | 0.099 ± 0.013     | 0.174 ± 0.009     |
> | rl-comm-with-biases (Eccles et al.) | 0.142 ± 0.019     | 0.729 ± 0.072     |
> | ae-comm (ours)                      | **0.348 ± 0.041** | **0.984 ± 0.002** |
>
>
> (All confidence intervals reported are 95% confidence intervals, as in Eccles et al.)
>
> We will continue to run experiments on the FindGoal environment, since those take longer time. We will update our response once the remaining experiments are completed, and will include all these results in the revised version of our paper.
>
> Although we did not have the time to do so yet, we plan to also compare our method to DIAL for the revision.
>
> **Ambiguity of figure 3c**
>
> We realize that there are three factors that may have contributed to the visual ambiguity in Figure 3c (i.e. large overlap of error regions):
> - The small number of runs we had (over 5 random seeds).
> - The small number of trials for each evaluation data sample (over 10 trials).
> - We reported the standard deviation of the moving average in each graph, rather than a 95% confidence interval as the shaded region (as in Eccles et al.).
>
> It is also worth noting that the FindGoal task can be solved without communication, unlike the other two tasks. The difference in agent performance is larger when there are more agents and/or there is a larger map. Given our limited computational resources and time constraints for the rebuttal, we found with our current settings to achieve the best balance between valid results and training time.
>
> During the past few days, we have been running additional experiments (5 more seeds) on the FindGoal environment, and evaluating on 100 trials per data point, to lessen this ambiguity. We have re-calculated the statistics and are reporting below the mean and 95% confidence intervals. We hope these demonstrate the stability and effectiveness of our approach.
>
> | average episode length | FindGoal         |
> |------------------------|------------------|
> | no-comm                | 169.0 ± 26.8     |
> | rl-comm                | 184.8 ± 25.2     |
> | ae-comm (ours)         | **103.5 ± 20.2** |
>
> We will continue to run more experiments and incorporate the results in the next revision. We will do the same for other environments as well, so that experiment results in our paper are standardized.
>
> **Positive listening**
>
> Thank you for pointing out the lack of scientific rigor here. We agree that Figure 6 does not sufficiently illustrate positive listening and will rework this section in the revised paper.
>
> Specifically, we will move Figure 6 to appendix and instead clarify the positive listening aspect with a table reporting increase in reward when adding a communication channel. This is suggested by [22] to be a sufficient metric for positive listening.
>
> | Δr                                  | CIFAR             | RedBlueDoors      |
> |-------------------------------------|-------------------|-------------------|
> | rl-comm                             | 0.017 ± 0.022     | -0.030 ± 0.155    |
> | rl-comm-with-biases (Eccles et al.) | 0.060 ± 0.029     | 0.552 ± 0.155     |
> | ae-comm (ours)                      | **0.266 ± 0.058** | **0.807 ± 0.185** |
>
>
> **Hypotheses about poor performance of ae-rl-comm baseline**
>
> Thank you for the suggestions! Our hypothesis is indeed that optimization is harder whenever the joint-exploration problem for learning speaker and listener policies is introduced. We do not have a solution yet, but our empirical results can confirm that the solution is not as simple as requiring more training iterations. We have run training to up to 1 million iterations and did not see an improvement in the result. We have also tried training the autoencoding component and RL component alternatingly, i.e. freezing one while training the other until loss stabilizes, yet performance degrades whenever the RL objective is optimized. We will include the training graphs and more discussion on this in our revised paper.
>
> **Discussion on limitations of the work**
>
> We will make sure to discuss the limitations of our work in more detail in our revised paper. In particular, our method indeed assumes that all agents have the same model architecture and the same autoencoding loss. We believe that testing the limits of this constraint, and relaxing it, will be important steps for future work, and will discuss this in the revision.
>
> **Suggested citations**
>
> Thank you for the suggestions. Indeed these are interesting works and we will cite them in our revised paper.
>
> **Small comments**
>
> We will update the terms to make them more precise and correct the typos in our revised paper.
>
> We also really appreciate the suggestion on additional references.
> - We will refer to Farrer 1987 for the definition of "cheap talk".
> - We have carefully read through the introduction section Dafoe et al.; we found it relevant and very enlightening, and will cite the discussion in our revised paper.
> - Though we have not had the time to read through "Origins of Human Communication" completely, we appreciate this great recommendation, and will incorporate a related discussion into our introduction if space permits.

---

> > ### Comment · Reviewer_ux54 · 2021-08-11
> > **Response to Authors**
> >
> > I am impressed by the authors' responses, particularly:
> >
> > (1) Their addition of the suggested missing baselines,
> > (2) Improvement in the statistical clarity of their results,
> > (3) Better discussion of how autoencoding sidesteps hard joint-exploration, and why adding RL on top of this is non-trivial,
> > (4) Better identification of limitations and how these might power future work.
> >
> > As a small additional suggestion, I would encourage the authors to identify the difficulty of adding RL on top of autoencoding as an important question for future work: this seems like a non-trivial problem from their results. It would certainly be a useful problem to solve, since then one will get more true "communication", where both speaker and listener have independent agency, based on the common "grounding" provided by autoencoding.
> >
> > Assuming that the authors are able to translate these improvements into the figures and text of the paper, I am happy to increase my review score (which I will do by editing my original review).

---

> > > ### Comment · Reviewer_ux54 · 2021-08-11
> > > **Follow-up question**
> > >
> > > As a follow-up question:
> > >
> > > Did the authors consider using a latent state from before the policy head as a communication vector? Do they expect that this would perform better than their fc-comm baseline, or not? The advantage that this might confer is that these latent states are easily accessible, and continuously trained with the policy reward, so it may be that they contain useful and relevant information. On the other hand, this information is perhaps more "privileged" than the fc-comm baseline because it is taken from deeper in the network than the policy layer, rather than from a layer on top of the policy.
> > >
> > > [I am taking my understanding of your fc-comm baseline from the discussion with reviewer rwT3 below; i.e. that this is an untrained fully connected layer on top of the policy. Please correct me if I have misunderstood this (and do please include a precise description of the setup in the final paper)].

---

> > > > ### Author Response · Authors · 2021-08-12
> > > > **Follow-up response to Reviewer ux54**
> > > >
> > > > Yes, your understanding of our "fc-comm" baseline is correct, and we will revise our description to make it more precise.
> > > >
> > > > We have indeed run experiments where agents transmit a latent state from before the policy head as a communication vector, and the performance was better than our "fc-comm" baseline (closer to "ae-comm"). We hypothesize that the reason for this is the same as what you mentioned: these latents states are continuously trained with the policy reward, and contain more useful and relevant information.
> > > >
> > > > If you feel that it would be helpful to add this baseline, we will include it in our revised paper as well.

---

> > > > > ### Comment · Reviewer_ux54 · 2021-08-13
> > > > > **Follow-up response to authors**
> > > > >
> > > > > It would certainly be helpful to add this baseline to the paper, to justify your claim that the auto-encoding is a necessary ingredient, over and above the transmission of an arbitrary latent state.
> > > > >
> > > > > Can you possibly respond with an updated version of the tables above which also include a line for this "latent-comm" baseline? I would like to satisfy myself that ae-comm significantly outperforms "latent-comm", since the importance of autoencoding is a central thesis in your work.

---

> > > > > > ### Author Response · Authors · 2021-08-17
> > > > > > **Follow-up response to Reviewer ux54**
> > > > > >
> > > > > > Here is an updated table that compares "ae-comm", "fc-comm" and "latent-comm" in two of our environments:
> > > > > >
> > > > > > | average reward | CIFAR             | RedBlueDoors      |
> > > > > > |----------------|-------------------|-------------------|
> > > > > > | fc-comm        | 0.107 ± 0.021     | 0.382 ± 0.022     |
> > > > > > | latent-comm-1        | 0.129 ± 0.034     | 0.714 ± 0.073     |
> > > > > > | latent-comm-2        | 0.092 ± 0.012     | 0.739 ± 0.080     |
> > > > > > | ae-comm (ours) | **0.348 ± 0.041** | **0.984 ± 0.002** |
> > > > > > | full-latent-comm        | 0.346 ± 0.070     | 0.912 ± 0.046     |
> > > > > >
> > > > > > FindGoal experiments take longer to run; we will update the results once they are completed. We will include the full baseline comparison results in our revised paper.
> > > > > >
> > > > > > For fair comparison, for all methods we restrict agents to use the same fixed-size discrete communication channel (except for "full-latent-comm", which we will explain in the next paragraph). The latent activation vectors in the policy nets are high-dimensional; reducing them to the size of the communication channel can be achieved in multiple ways. In "latent-comm-1", we do so by reducing the size of the last hidden layer of the policy net so that it matches the size of the communication channel. In "latent-comm-2", we instead use the original policy architecture and only transmit the first N features of the last hidden layer of the policy net, where N is the size of the communication channel. In all cases, we quantize the outputs identically and use a straight-through estimator to differentiate through the quantization.
> > > > > >
> > > > > > As an "upper bound", we also compare all methods to "full-latent-comm", a method that communicates the full, high-dimensional and continuous latent vector from the last hidden layer of the policy net. As shown in the table, the performance of "full-latent-comm" is close to that of "ae-comm". Clearly, if agents can directly observe each other’s full internal states, they can coordinate their behavior well. Our contribution is to show that similar performance can be achieved even when agents are only allowed to make low-dimensional and discrete utterances -- a setting that more closely models communication in nature and may have greater practical utility (e.g., low-bandwidth communication channels between robots).

---

> > > > > > > ### Author Response · Authors · 2021-08-17
> > > > > > > **Remaining results for Eccles et al.**
> > > > > > >
> > > > > > > Here are the remaining results for Eccles et al.:
> > > > > > >
> > > > > > >
> > > > > > > | average episode length | FindGoal         |
> > > > > > > |------------------------|------------------|
> > > > > > > | no-comm                | 169.0 ± 26.8     |
> > > > > > > | rl-comm                | 184.8 ± 25.2     |
> > > > > > > | rl-comm-with-biases (Eccles et al.) | 157.99 ± 12.3     |
> > > > > > > | ae-comm (ours)         | **103.5 ± 20.2** |
> > > > > > >
> > > > > > >
> > > > > > > We will include the full results in our revised paper.

---

> > > > > > > ### Comment · Reviewer_ux54 · 2021-08-18
> > > > > > > **Response to Authors**
> > > > > > >
> > > > > > > Many thanks for these results. I find the method and the argument convincing. Please include these in the paper, particularly the "full-latent-comm" topline. That will help to clarify that your contribution lies in a method for communicating in discrete and low-dimensional utterances, rather than in aligning the latent states of the agents.

---

> > > > > > > > ### Author Response · Authors · 2021-08-18
> > > > > > > > **Thank you**
> > > > > > > >
> > > > > > > > Thanks for all your comments and quick responses! We will include these results in the revised paper and clarify the contribution.

---

> > > ### Author Response · Authors · 2021-08-12
> > > **Follow-up response to Reviewer ux54**
> > >
> > > Thank you for the positive feedback.
> > >
> > > We fully agree that making RL work on top of autoencoding is a difficult yet essential problem to solve. Our revision will highlight its difficulty alongside empirical results demonstrating the complexity of training emergent communication via policy optimization.

---

### Author Response · Authors · 2021-08-10
**Thank you for the review**

We thank all the reviewers for their extremely helpful feedback. Below, we respond to each reviewer individually. Please feel free to post additional comments if further clarification is needed.

---

### Decision · Program_Chairs · 2021-09-27

**Decision:**

Accept (Poster)

**Comment:**

Reviewers found the proposed method a simple yet intriguing contribution to the emergent communication literature. The authors addressed important initial concerns about baselines and ablations in the rebuttal period, and reviewers were satisfied with the new results. There was concern about whether sending autoencoded latents "really" constitutes communication, however I believe that makes the work all the more thought-provoking for the community. I expect this paper will be a useful contribution for other emergent communication research to use as a baseline or build upon.